# Preparation and Characterization of Pickering Emulsions with Modified Okara Insoluble Dietary Fiber

**DOI:** 10.3390/foods10122982

**Published:** 2021-12-03

**Authors:** Yue Bao, Hanyu Xue, Yang Yue, Xiujuan Wang, Hansong Yu, Chunhong Piao

**Affiliations:** 1College of Food Science and Engineering, Jilin Agricultural University, Changchun 130118, China; baoyue0407@163.com (Y.B.); xuehanyu0226@163.com (H.X.); 15943188314@163.com (Y.Y.); juan4420@163.com (X.W.); yuhansong@jlau.edu.cn (H.Y.); 2National Engineering Laboratory for Wheat and Corn Deep Processing, Changchun 130118, China

**Keywords:** microstructure, okara insoluble dietary fiber, pickering emulsions, stability, yeast *Kluyveromyces marxianus*

## Abstract

Modified okara insoluble dietary fiber (OIDF) has attracted great interest as a promising Pickering emulsifier. At present, the modification methods are mainly physicochemical methods, and the research on microbial modified OIDF as stabilizer is not clear. In this work, modified OIDF was prepared by yeast *Kluyveromyces marxianus* fermentation. The potential of modified OIDF as a Pickering emulsifier and the formation and stability of OIDF-Pickering emulsions stabilized by modified OIDF were characterized, respectively. The results showed that the specific surface area, hydrophilicity, and electronegativity of the modified OIDF were all enhanced compared with the unmodified OIDF. The existence of the network structure between droplets is the key to maintain the stability of the emulsions, as indicated by Croy-Scanning Electron Microscope (Croy-SEM) and rheological properties measurements. The stability of OIDF-Pickering emulsions was evaluated in terms of storage time, centrifugal force, pH value, and ionic strength (NaCl). Moreover, the OIDF-Pickering emulsions stabilized by modified OIDF showed better stability. These results will contribute to the development of efficient OIDF-based emulsifiers, expand the application of emulsions in more fields, and will greatly improve the high-value utilization of okara by-products.

## 1. Introduction

Pickering emulsion is a novel emulsion stabilized by amphiphilic solid particles [1]. The key to its stability is the irreversible adsorption of solid particles and the formation of a space barrier on the interface [2]. Recently, three main categories of biocompatible polymeric particles act as emulsifiers to stabilize Pickering emulsions have been revealed which are based on proteins, polysaccharides, and composite particles, respectively. The proteins-based particles include surimi particles [3], heat-treated tea water-insoluble protein nanoparticles [4], and myofibrillar microgel particles [5]. The polysaccharides-based particles include octenyl succinic anhydride (OSA)-modified cellulose nanocrystals [6], physical or chemical modified nanostarch particles [7], enzyme-treated banana peels cellulose nanofibers [8], and alkali-treated or ultrasonic-treated soybean insoluble fiber particles [9]. The composite-based particles include zein-tea saponin composite nanoparticles [10], dihydromyricetin-high-amylose corn starch composite particles [11], and zein-cellulose nanocrystals [10]. Wherein, polysaccharides-based particles have become a popular material due to their amphiphilic, natural aggregation characteristics and their abundant existence in nature.

Soybean (*Glycine max* (Linn.) Merr.) is one of the important grain crops worldwide, which is commonly used to make various soybean products, extract soybean oil, brew soy sauce, and extract protein [12]. Every year, about 70 million tons of okara is obtained after soybean processing, which is used as animal feed or discarded as waste due to additional processing costs [13]. However, numerous data show that the content of dietary fiber in the dry matter of okara is up to 55%, of which the content of insoluble dietary fiber (IDF) is as high as 90%. It is considered as an excellent natural dietary fiber and has gradually attracted more attention [14,15]. In terms of raw okara, the dense structure limited the extraction and application of okara insoluble dietary fiber (OIDF). In recent years, some studies have focused on the structural characteristics, physicochemical properties, and emulsification properties of OIDF. These studies have proved that OIDF can be used as a Pickering emulsifier if subjected to certain modification treatments [16], opening up a new way for the high-value utilization of okara. Modification treatment mainly relies on physicochemical methods, such as high-energy wet media grinding [17], alkali treatment, ultrasonic alkali treatment, cooking alkali treatment, ultrasonic-assisted cooking alkali treatment [9], or other modification methods. Meanwhile, physicochemical methods are often accompanied by high energy consumption, the potential hazards of environmental pollution, and so on.

Microbial fermentation modified okara is also one of the common modification methods. Frequently used microbials were probiotics, yeast, and their composite fermentation system, and the qualities related to okara processing were significantly improved by fermentation. In our previous study, yeast *K. marxianus*, which has obtained the EU and US safety qualification (QPS) and GRAS status and has been certified as food-grade yeast [18], was used to modified okara to prepare OIDF. It was found that the modified OIDF exhibited a looser and porous structure, higher porosity, emulsification characteristics, as well as a significant improving effect on the water-holding capacity and oil-holding capacity [14,19]. 

Therefore, in this work, OIDF modified by microbial fermentation was applied to the field of Pickering emulsions for the first time. Pickering emulsions prepared using modified OIDF by yeast *K. marxianus* fermentation were compared with using unmodified OIDF, and the stability of those OIDF-Pickering emulsions were also characterized. It is believed that the research can provide very good experimental data for extending the application of okara as emulsifiers.

## 2. Materials and Methods

### 2.1. Materials 

Soybeans (Heihe 43, Heihe, China) were purchased from Shandong Shengfeng Seeds Co., Ltd. (Zibo, China) Yeast *Kluyveromyces marxianus* were isolated from kefir grains and deposited at the China General Microbiological Culture Collection Center (CGMCC, Beijing, China) under accession number 13907. Soybean oil (grade A) was purchased from Jiusan Grain and Oil Industry Group Co., Ltd. (Harbin, China). Sudan was purchased from Beijing Dingguo Biotechnology Co., Ltd. (Beijing, China). All other chemicals used were of analytical grade.

### 2.2. Preparation of Modified OIDF 

Modified OIDF was prepared from okara fermentation by yeast *K. marxianus* based on our previous method [19]. The okara was soaked in distilled water at a ratio of 1:5 (*w*/*v*) and sterilized at 121 °C for 20 min in a vertical pressure steam sterilization pot (YXQ-S-50A, Shanghai Boxun Enterprise Co., Ltd., Shanghai, China). The sterilized okara was inoculated with *K. marxianus* powder at a ratio of 10% (*w*/*v*) and fermented for 72 h. Raw OIDF without any modified treatment was used as a control. All samples were freeze-dried by vacuum freeze dryer (FDU-7006, Oberon Co., Ltd., Gimpo, Korea), and crushed using an ultramicro pulverizer (Youqi Co., Ltd., Osaka, Japan) for 2 min, stored in the dryer (Transparent version 180 mm, Fangge Medical Instrument Co., Ltd., Beijing, China). 

The content of IDF in raw and modified OIDF were determined to 70.53% and 72.38% according to national standards GB 5009.88-2014, and the content of crude protein, crude lipid, moisture and ash in raw and modified OIDF were determined to 70.53%, 12.0%, 5.42%, 8.60%, 1.31% and 72.38%, 9.90%, 8.41%, 5.56%, 1.02% according to national standards GB 5009-2016.

### 2.3. Preparation of Pickering Emulsions with Modified Okara Insoluble Dietary Fiber

The OIDF suspensions containing 0, 0.4, 0.8, 1.0, 1.2, 1.6, and 2.0 wt% OIDF powder were mixed by a homogenizer (Tissue-Tearor, BioSpec, Rockville, MD, USA) at 10,000 rpm, for three times and each time for 3 min, with an interval time of 30 s. And then operated with an ultrasonic cell grinder (Ningbo Xinzhi Biotechnology Co., Ltd., Ningbo, China) at an ice bath at 500 W for 3 s and an interval of 3 s for 30 min. 

The OIDF-Pickering emulsions, containing soybean oil and OIDF suspensions, were emulsified according to the two-step method (shear and ultrasonication) proposed by Bai et al. [20]. Firstly, OIDF-Pickering emulsions were homogenized by a homogenizer (Fluke Fluid Machinery Manufacturing Co., Ltd., Everett, WA, USA) at 13,000 rpm for 2 min, then immediately operated with an ultrasonic cell grinder (Ningbo Xinzhi Biotechnology Co., Ltd., Ningbo, China) at 500 W for 3 s and an interval of 3 s for 3 min. The optimization of OIDF-Pickering emulsion preparation formulation was achieved by varying the ratio of soybean oil and OIDF suspensions, specifically produced 11 different treatment combinations (Table 1).

### 2.4. Particle Size and Zeta Potential Analysis 

The particle size of OIDF and its stable OIDF-Pickering emulsions was measured by a laser particle size analyzer (BT-9300HT, Baidu Instrument Co., Ltd., Dandong, China). Distilled water was used as a dispersant, and the refractive index of the water was 1.33 and the background intensity of the shade was 14%. The OIDF suspensions were added to the sample cell equilibrated at 25 °C for 2 min, and then measured the zeta potential using a zeta potential analyzer (Litesizer 500, Anton Paar (Shanghai) Co., Ltd., Graz, Austria). 

### 2.5. Contact Angle Measurement

The contact angles of the samples were measured according to the method of Chen et al. [6] using KINO static and dynamic contact angle meter (C601, Kino Industry Co., Ltd., Boston, MA, USA). The OIDF powders were pressed into coin-sized tablets (20 mm in diameter and 2 mm in height) and placed on the glass slide. Then, the surface was covered with soybean oil and transferred to the sample table. A droplet of distilled water (2 µL) was injected onto the surface of tablets using a high-precision injector. After equilibration for 5 s, the droplet image was recorded by a high-speed camera. The profile of the droplet was numerically analyzed and automatically fitting to the Laplace Young equation to obtain a contact angle.

### 2.6. Morphological Observation of OIDF-Pickering Emulsions

The morphology of OIDF-Pickering emulsions was observed using an inverted microscope (Leica DMIL LED, Leica Group Co., Ltd., Dresden, Germany). Before that, the type of emulsions was determined according to the principle of “like dissolves like” [3]. OIDF-Pickering emulsions were stained with Sudan (5 mg/mL, dissolved in 95% ethanol) at a ratio of 2:1 (*v*/*v*), 20-µL emulsions were placed onto the glass slide, and the glass was gently covered to avoid air bubbles. The magnification was adjusted to 50×.

### 2.7. Croy-Scanning Electron Microscope (Croy-SEM)

A cryo-scanning electron microscope (Zeiss Sigma 300, Carl Zeiss AG, Munich, Germany) was used to observe the OIDF-Pickering emulsions for microstructure. The fresh samples were pipetted onto a copper sample plate, quickly frozen in liquid nitrogen, and transferred into the croy-preparation chamber under vacuum. The sample was sublimated for 15 min to remove part of the water, then cut into a flat cross section and sprayed with gold, and observed under an electron microscope.

### 2.8. Rheological Properties

The rheological properties of OIDF-Pickering emulsions were measured using a rheometer (MCR302, Austria Anton para Co., Ltd., Graz, Austria) at room atmosphere. The steady shear scanning was conducted at a shear of 0.1–100 s^−1^ using a flat plate (25 mm in diameter) under the dynamic and static shear program [21]. The power law model and Herschel–Bulkley model were fitted based on the following Equations (1) and (2), respectively.
(1)$${\eta =\kappa \gamma }^{n-1}$$
(2)$${\tau =\tau }_{0}+\kappa \gamma^{a}$$
where η is the viscosity (Pa·s^−1^) and γ is the shear rate (s^−1^); κ is the consistency coefficient, which is the intrinsic viscosity of the system at a shear rate of 1 s^−1^; n is the power law index; τ is the shear stress (Pa); $\tau_{0}$ is the yield stress ; γ is the shear rate (s^−1^); and κ and a are constants in the model.

### 2.9. Stability Analysis of OIDF-Pickering Emulsions

#### 2.9.1. Storage Stability

To evaluate the storage stability, OIDF-Pickering emulsions were placed into the 25 mL of collecting bottles at room temperature for 28 days. The emulsified phase volume fraction (EPVF) of the samples was calculated using the following Equation (3), according to the method in the previous publication [3].
(3)$$\mathrm{EPVF}(\%)=\frac{H_{e}}{H_{t}}\;\times \;100\%$$
where H_e_ stands for the heights of emulsion phase, and H_t_ stands for the total heights of emulsion system.

#### 2.9.2. Centrifugal Stability (Encapsulation Efficiency)

The centrifugation stability of OIDF-Pickering emulsions was characterized by calculating the encapsulation efficiency (EE) of the samples according to the previous publication with some modifications [22]. This experiment used n-hexane as the extractant to wash free soybean oil. The samples were mixed with n-hexane at a ratio of 1:2 (*v*/*v*) and placed in a centrifuge (TGL-16G, Shanghai fichar Analytical Instrument Co., Ltd., Shanghai, China) and centrifuged at different speeds (0, 12,000, 24,000, 36,000 *g*) for 10 min. The OD value of supernatant was measured at the optimal absorption wavelength (295 nm), where scanned soybean oil with a full wavelength (200–1000 nm) was used to determine the optimal absorption wavelength. The encapsulation efficiency was calculated (4) according to the following equation.
(4)$$\mathrm{EE}(\%)=\frac{{O/W}_{t}-{O/W}_{f}}{{O/W}_{t}}\;\times \;100\%$$
where O/W_t_ and O/W_f_ refer to the total and free oil–water ratio, respectively. O/W_t_ and O/W_f_ were calculated according to the standard curve (Y = 3.6445X + 0.895, R^2^ = 0.9962) with different oil–water ratios changes from 0:10 to 6:4 (*v*/*v*) as abscissa and OD value as ordinate. Then, 200 µL of n-hexane was added to the different ratios of oil–water mixtures whose total volume was 300 µL when the OD value was obtained.

#### 2.9.3. pH Value or Ionic Strength Stability

To evaluate the pH value or ionic strength (NaCl) stability of OIDF-Pickering emulsions, the pH value was adjusted to pH 3, pH 5, pH 7, pH 9, and pH 11, and the ionic strength (NaCl) was adjusted to 0, 100, 200, 300, 400, and 500 mM, respectively. Then, the adjusted samples were immediately and magnetically stirred at 25 °C for 1 h [23]. Subsequently, the zeta potential of the adjusted samples was measured after storage at 25 °C for 72 h.

### 2.10. Statistical Analysis

All data were obtained from triplicate trials and are expressed as the average value ± SD. GraphPad prism 8.0 (GraphPad Software Inc., San Diego, CA, USA) and Origin lab origin 2019 software (Microcal Inc., Northampton, MA, USA) were used to draw test curves, and IBM SPSS 22 software (SPSS Inc., Chicago, IL, USA) was used to analyze significant differences. Differences were considered to be significant when *p* < 0.05 using Fisher’s least significant difference (LSD) test. Microsoft Excel 2000 software (Microsoft Inc., Redmond, WA, USA) was used to draw the test data of the standard curves.

## 3. Results

### 3.1. Basic Characteristics of OIDF

The study of OIDF revealed that modified OIDF by yeast *K. marxianus* led to a decrease in the polymerization degree of cellulose and a looseness of the structure [19]. In addition, the formation and stability of Pickering emulsion are closely related to the morphology and properties of solid particles [24]. Therefore, the particle size, wettability, and zeta potential of OIDF were investigated (Figure 1). As represented in Figure 1A, the D50 of unmodified and modified OIDF were 41.86 ± 1.13 and 42.86 ± 2.07 µm, respectively, both showing single peak distribution due to the ultramicro treatment. Moreover, the specific surface area of the modified OIDF increased significantly by 22.8% than unmodified, which is beneficial to improve the surface coverage of emulsion [25].

The wettability of solid particles is related to the interfacial interaction between particles (adsorption), which determines the type of emulsions [26]. From the perspective of thermodynamics, as long as the *θ* is not close to 0° or 180°, the free energy of spontaneous desorption of particles is much greater than the thermal energy, then the adsorption of particles at the interface is irreversible, and the free energy of spontaneous desorption is the largest when the *θ* is 90°, when the adsorption is the most stable [27]. The *θ_O/W_* of unmodified and modified OIDF were 63.29 ± 2.39° and 49.65 ± 2.16°, respectively (Figure 1B), indicating enhanced hydrophilicity of modified OIDF which is related to the high water-holding capacity [19]. Cai et al. [16] found that the *θ_O/W_* reduction in insoluble soybean fiber is related to an increase in the water-holding capacity after alkaline treatment.

Moreover, the zeta potential is also one of the effective indices for evaluating the strength of electrostatic interaction. The higher the absolute value of zeta potential meant the stronger the electrostatic interaction, which is conducive to the stability of emulsions [21]. As shown in Figure 1C, the absolute value of the zeta potential of modified OIDF had a significant increase (*p* < 0.01), mainly because more surface charges were exposed after modification, caused by loose structure [19]. Overall, combined with the results of particle size, wettability, and zeta potential, the modified OIDF would be a great potential particle to stabilize Pickering emulsions.

### 3.2. Effects of O/W and OIDF Concentration on the Formation of OIDF-Pickering Emulsions

According to the characteristics of emulsion formation, the oil–water ratio and the change in solid particles fulfil a crucial role in the size of emulsion droplets and the stability of emulsions [15]. In this study, through visual, morphological observations and particle size analysis, the preparation conditions of OIDF-Pickering emulsions were carried out (Figure 2). It was observed that Pickering emulsions after standing for 24 h were stabilized eventually at different oil–water ratios, whether unmodified or modified OIDF, and the appearance of the modified OIDF emulsion phase was creamier than that of unmodified OIDF. When the oil–water ratio was 5:5 or more, clear oil precipitation was observed (Figure 2A left). Further morphological observation (Figure 2A right) showed that an enormous number of emulsion droplets were formed in the emulsion system. With the increase in the oil–water ratio, the emulsion droplets gradually appeared irregular. It was observed that the emulsion droplets were smaller at a 2:8 oil–water ratio due to a reduced oil phase in the system, indicating that OIDF was sufficient to stabilize droplets. Whereas, the emulsion droplets were gradually distributed uniformly and appeared regularly at a 4:6 oil–water ratio. When the oil–water ratio was 5:5 or more, OIDF was insufficient to stabilize the oil droplets. To reduce the interface area, the droplets coalesced, and the coalesced oil droplets adhered to the surface of the vial, resulting in oil precipitation. Therefore, the oil–water ratio of 4:6 was selected for the subsequent experiment, according to the requirement of the high oil phase. In similar studies, cellulose nanocrystals were used to stabilize sunflower seed oil at the oil–water ratio of 1:9 [28]. The selection of the oil–water ratio may be related to the particle size and the surface characteristics of solid particles [3].

As exhibited in Figure 2B, the concentrations of OIDF affected the formation of emulsions after standing for 72 h. Increasing the concentrations of OIDF from 0 wt% to 2.0 wt% gave rise to a significant increase in the emulsified phase volume, and the emulsified phase volume in modified OIDF was higher than that of unmodified OIDF. Yang et al. [15] also found that, following the increased concentration of insoluble okara polysaccharides from 0.25 wt% to 1.0 wt%, these particles with stabilized emulsions produced a higher emulsified phase volume and became more stable. At the same, the emulsion droplets gradually appeared more uniform and regular with the increase in the concentrations of OIDF from morphological observations. Bimodal distribution appeared in particle size analysis (Figure 2C), and the nanofiber stabilized Pickering emulsions also found a similar bimodal distribution [29]. The first peak should be attributed to the emulsion droplets in the system, and the second peak should be attributed to the unabsorbed OIDF in the system. With the increase in OIDF concentrations, the ratio of the first peak increased and the ratio of the second peak decreased, indicating that the volume of the emulsified phase increases and the amount of unabsorbed OIDF decreases. Besides, the particle size distribution of the OIDF-Pickering emulsions stabilized by modified OIDF were significantly higher than that stabilized by unmodified OIDF. Notably, the largest volume of emulsified phase and the least amount of unabsorbed OIDF were investigated at the concentration of modified OIDF of 0.8 wt%. In similar studies, Cai et al. [16] used soybean dregs insoluble dietary fiber at a concentration of 1.0 wt% to stabilize the O/W emulsions, and Bai et al. [30] used a concentration of 2.0 wt% nanocellulose to stabilize the O/W emulsions. Finally, the OIDF-Pickering emulsions were prepared under an oil–water ratio of 4:6, an OIDF concentration of 0.8 wt%, and 500-W ultrasonic treatment for 6 min, and the ultrasonic treatment condition was optimized (Appendix A).

### 3.3. Characteristics of OIDF-Pickering Emulsions 

#### 3.3.1. Basic Physical Characteristics of OIDF-Pickering Emulsions

According to the dispersion behaviors of OIDF-Pickering emulsions in the oil–water phase, it was found that the droplets dispersed quickly in the water phase and remained stable in the oil phase, which is consistent with the characteristics of O/W emulsions, in line with the contact angle results (Figure 3A). The zeta potential of OIDF-Pickering emulsions stabilized by unmodified and modified OIDF were −10.56 ± 1.41 and −31.99 ± 1.34 mV, respectively (Figure 3B). Particle size distribution and morphological analysis well expressed the state of the emulsion droplets. As shown in Figure 3C, OIDF-Pickering emulsions exhibited bimodal distribution but the OIDF-Pickering emulsions stabilized by modified OIDF were more homogeneous. As seen from the insert optical images, it was also found that the OIDF-Pickering emulsions stabilized by unmodified OIDF have coalesced. A similar coalescence phenomenon which occurred in the Pickering emulsions by combined cellulose nanofibrils and nanocrystals [31] is mainly due to the weak electrostatic interaction between the emulsions [26] or the difference in the surface properties between the unmodified and modified OIDF. 

#### 3.3.2. Croy-SEM of OIDF-Pickering Emulsions

Croy-SEM technology includes rapid freezing treatment of samples and scanning electron microscopy, which can directly observe liquid and semi-liquid. Wherein, rapid freezing technology can make water glassy at a low temperature and reduce the generation of ice crystals, so as not to affect the structure of the sample itself. To reveal emulsion droplets and oil–water interface, the microstructure of the OIDF-Pickering emulsion was observed using Croy-SEM technology (Figure 4). From the Croy-SEM images, the formation of the network structure between emulsion droplets–droplets can be observed clearly. The network structure was established as a space barrier by OIDF, which prevented emulsion droplets from coalesce and maintained the stability of the emulsions. Yuan et al. [32] also found the formation of a 3D network structure of droplet fibers in the Pickering emulsion system stabilized by cellulose fibers. Meanwhile, it is also clearly observed that the network structure of the OIDF-Pickering emulsion stabilized by modified OIDF showed more loosening, while the network structure of the OIDF-Pickering emulsion stabilized by the unmodified OIDF showed more densely (Figure 4B), which is related to the higher porosity and lower bulk density of the modified OIDF after the modification treatment [19]. For Pickering emulsions, the excessive accumulation of network structure caused strong flocculation and falling off, which resulted in coalescence and gravity separation of the emulsions [33]. In conclusion, Croy-SEM images proved that modified OIDF could be used as a Pickering emulsifier to stabilize Pickering emulsions, and the network structure formed between droplets is the key to maintaining the stability of the emulsions.

#### 3.3.3. Rheological Properties of OIDF-Pickering Emulsions

Rheological properties reveal the droplet–droplet and solid particles–solid particles interactions in the emulsion systems [34]. Herein, to investigate the stability mechanism, the rheological parameters of OIDF-Pickering emulsions were measured (Figure 5). The power law model was carried out to fit the viscosity–shear rate relationship to characterize the droplet–droplet interaction. As shown in Figure 5A, OIDF-Pickering emulsions displayed the shear-thinning behavior, indicating the presence of weak interaction between the droplets. It further proved the existence of the network structure [8]. The reason for the decrease in viscosity was that the broken network structure and the solid particles were aligned along the flow direction under the action of shear force. Furthermore, n ˂ 1 in the power law model also confirmed the shear-thinning characteristic. The consistency coefficients (κ) were 3.31 and 5.67 in the OIDF-Pickering emulsions stabilized by unmodified and modified OIDF, respectively. Meanwhile, the viscosity of OIDF-Pickering emulsions stabilized by modified OIDF was always higher than that of stabilized by unmodified OIDF during the whole shearing process, indicating the network structure of OIDF-Pickering emulsions stabilized by modified OIDF was superior to the OIDF-Pickering emulsions stabilized by unmodified OIDF in resisting the shear force [35].

To explain this phenomenon, we have to consider the solid particles–solid particles interaction in the network structure. The Herschel–Bulkley model was carried out to fit the shear stress–shear rate relationship. For any system with a network structure, the initial shear stress will cause the deformation of the network structure. When the applied stress is greater than the yield stress, the emulsion begins to flow, so the yield stress ($\tau_{0}$) can be used to predict the stability of the network structure in the emulsions [36]. As shown in Figure 5B, the yield stress ($\tau_{0}$) of OIDF-Pickering emulsions stabilized by unmodified and modified OIDF were 3.16 and 4.77 Pa, respectively, which were higher than that of different morphologies of cellulosic fibers, including extruded cellulose, incomplete nano fibrotic cellulose of 0.28 Pa, incomplete nano fibrotic cellulose of 0.96 Pa, and complete nano fibrotic cellulose of 2.90 Pa [32]. The consequence is mainly attributed to higher porosity, resulting in dispersion of resistance to shear stress and electrostatic repulsion force between modified OIDF, as well as increased stability in the network structure [22]. In short, the rheological results showed that modified OIDF had the capacity to form a more stable network structure compared to unmodified OIDF, which can effectively resist the coalescence of droplets to improve the stability of the emulsion.

### 3.4. Stability Analysis of OIDF-Pickering Emulsions

#### 3.4.1. Storage Stability

In this study, OIDF-Pickering emulsions were stored for 28 days. As shown in Figure 6, during the 28-days storage period, no obvious stratification occurred in OIDF-Pickering stabilized by modified OIDF. However, the OIDF-Pickering emulsions stabilized by unmodified OIDF began to appear delamination on the 7th day, and then the phenomenon was extended more seriously with the increase in the storage period. The volume fraction (EPVF) was provided in Table 2, which further certificated the results of emulsion observation. On the 7th day, the EPVF of OIDF-Pickering emulsions stabilized by modified OIDF was 100%, while the EPVF of OIDF-Pickering emulsions stabilized by unmodified OIDF was 91% and the bottom phase performed the aqueous phase. Until the 28th day, the EPVF of OIDF-Pickering emulsions stabilized by unmodified OIDF was 78%, and the EPVF of the OIDF-Pickering emulsion stabilized by the modified OIDF only decreased to 95%, while the bottom phase still performed Gel-like emulsion. Similarly, the EPVF of the Pickering emulsions stabilized by surimi particles was only 98% on the 14th day [3]. As expected, the OIDF-Pickering emulsions stabilized by modified OIDF showed the strongest storage stability, due to the homogeneous droplet distribution and strong electrostatic interaction (Figure 3). The existence of network structure of emulsion droplets (Figure 4) and higher yield stress prove the stability of network structure (Figure 5).

#### 3.4.2. Centrifugal Stability (Encapsulation Efficiency)

Centrifugal treatment aggravated the process of emulsion stratification, which further judged the stability of emulsions. Here, the changes in encapsulation efficiency of OIDF-Pickering emulsions under different centrifugal force (12,000, 24,000, 36,000 *g*) were measured. 

As exhibited in Figure 7A, under different speeds, OIDF-Pickering emulsions stabilized by unmodified and modified OIDF showed distinct differences, and the encapsulation efficiency of OIDF-Pickering emulsions stabilized by modified OIDF was always higher than that of stabilized by unmodified OIDF, which was consistent with the results of storage stability. The encapsulation efficiency of OIDF-Pickering emulsions without centrifugal treatment and stabilized by modified OIDF was about 95%, which was higher than that of stabilized by unmodified OIDF of 80%, mainly due to the loose structure and the large porosity of modified OIDF (Figure 2). Combined with a larger surface area of modified OIDF (Figure 1) and more stable emulsions (Figure 5), the OIDF-Pickering emulsions in potential application to the delivery of bioactive substances is greatly prospective.

#### 3.4.3. pH and Ionic Strength (NaCl) Stability 

The pH and ionic strength (NaCl) are elements in the emulsion preparation and application process. The original pH value of OIDF-Pickering emulsions was 5, so the pH 5 was selected as the control. Under different pH value (pH 3, pH 5, pH 7, pH 9, pH 11) conditions, unmodified OIDF-Pickering emulsions stabilized by unmodified and modified OIDF showed differences, indicating that different pH conditions have an effect on the surface charge distribution of the emulsion droplets. The OIDF-Pickering emulsions had a significant impact from pH 3 to pH 7. The absolute value of zeta potential increased slightly during pH 7–11, attributed to the deprotonation of some of the protonated carboxyl groups (−COOH → −COO^−^ + H^+^), released more charge [2]. Under ionic strength (NaCl, 0, 100, 200, 300, 400, 500 mM) conditions, OIDF-Pickering emulsions stabilized by unmodified and modified OIDF showed significant differences, owing to the electrostatic shielding effect, and the addition of ions will shield the surface charge of the emulsion droplets [2]. Moreover, the absolute value of zeta potential of OIDF-Pickering emulsions stabilized by modified OIDF was always greater than that stabilized by unmodified OIDF. In short, the excellent stability of OIDF-Pickering emulsions stabilized by modified OIDF will help expand its application in many fields.

## 4. Conclusions

In this work, we demonstrated that OIDF prepared by yeast *K. marxianus* fermentation can be regarded as Pickering emulsifiers for the preparation of OIDF-Pickering emulsions. Moreover, the OIDF-Pickering emulsions stabilized by the modified OIDF exhibited smaller droplets, stronger electrostatic interaction, a clearer and looser network structure, as well as higher consistency coefficient and yield stress. Moreover, due to the electrostatic interaction, the stability of OIDF-Pickering emulsions decreased under extreme pH and ionic strength (NaCl), but OIDF-Pickering emulsions stabilized by modified OIDF still exhibited certain pH or ionic strength (NaCl) adaptability. This work provided new insights for the okara insoluble dietary fiber-based Pickering emulsifier.

## Figures and Tables

**Figure 1 foods-10-02982-f001:**
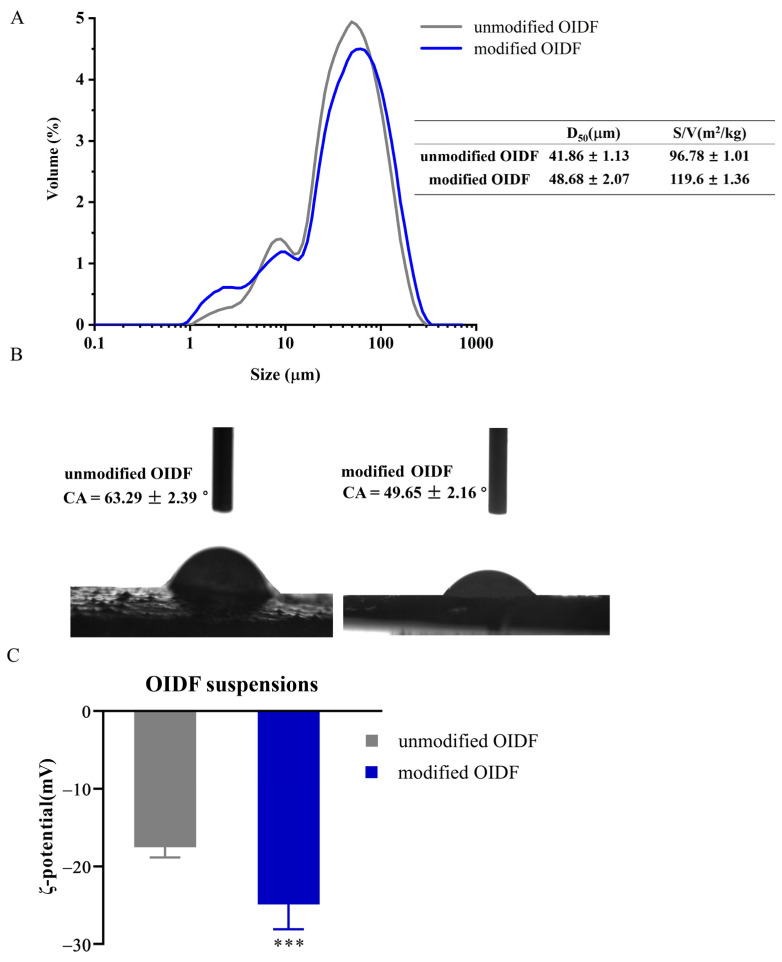
Basic characteristics of OIDF. (**A**) The size of OIDF; (**B**) The contact angle (CA) of OIDF; (**C**) The zeta-potential of OIDF (*** *p* < 0.001).

**Figure 2 foods-10-02982-f002:**
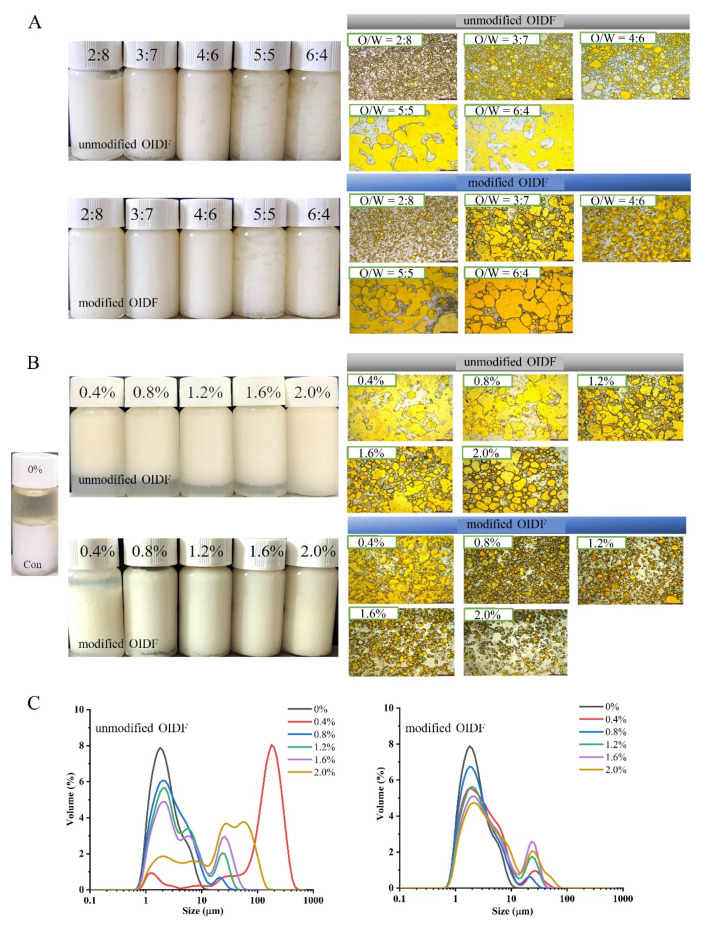
The effect of O/W and OIDF concentration on the formation of OIDF-Pickering emulsions. (**A**) Visual (left) and morphological (right) observation of OIDF-Pickering emulsions at different O/W after standing for 24 h; (**B**) Visual (left) and morphological (right) observation of OIDF-Pickering emulsions with different OIDF concentrations after standing for 72 h; (**C**) Size distribution of OIDF-Pickering emulsions with different OIDF concentrations.

**Figure 3 foods-10-02982-f003:**
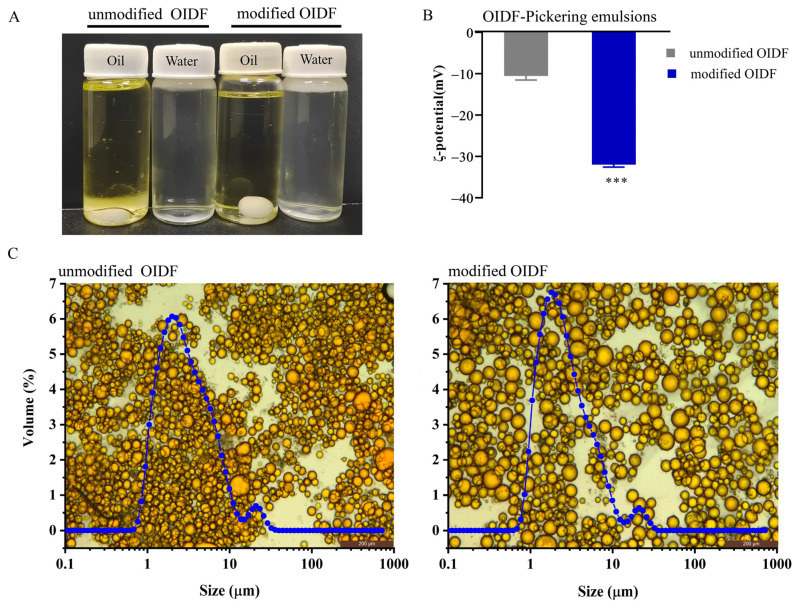
Basic physical characteristics of OIDF-Pickering emulsions. (**A**) The dispersion characteristics of OIDF-Pickering emulsions; (**B**) The zeta potential of OIDF-Pickering emulsions; (**C**) Size distribution and morphological observation of OIDF-Pickering emulsions (*** *p* < 0.001).

**Figure 4 foods-10-02982-f004:**
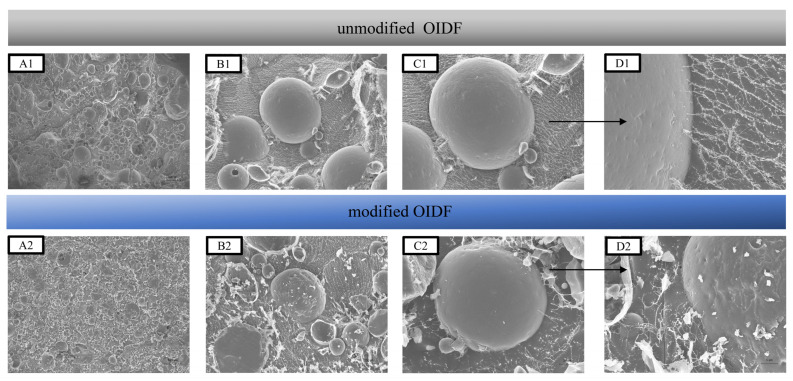
Cryo-SEM images of OIDF-Pickering emulsions. (**A1**,**A2**) The overall appearance of the emulsion droplets; (**B1**,**B2**) The microstructure of the emulsion droplets; (**C1**,**C2**) The interface structure of a single emulsion droplets; (**D1**,**D2**) The microstructure of the emulsions at the interface.

**Figure 5 foods-10-02982-f005:**
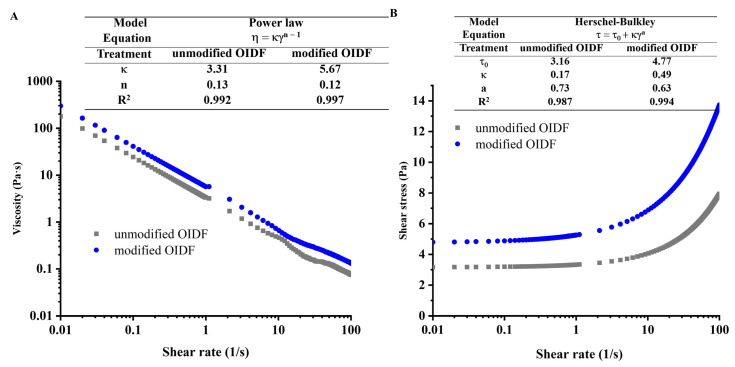
Rheological properties of OIDF-Pickering emulsions. (**A**) Viscosity–shear rate relationship of OIDF-Pickering emulsions; (**B**) Shear stress–shear rate relationship of OIDF-Pickering emulsions.

**Figure 6 foods-10-02982-f006:**
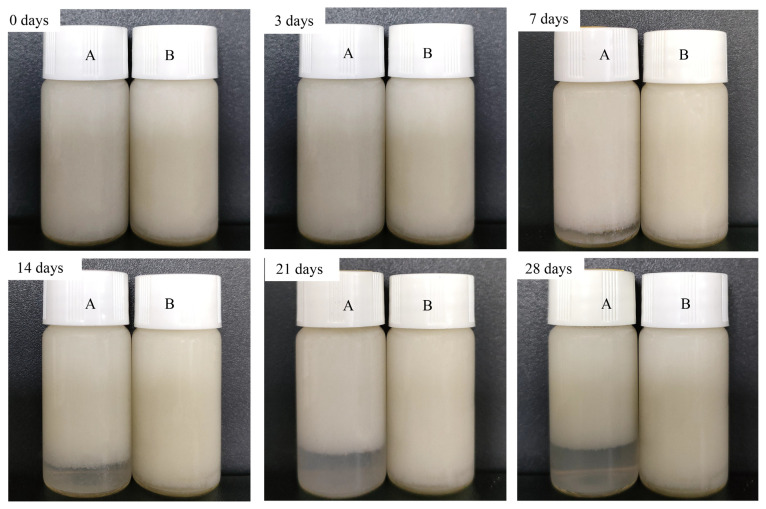
Storage stability of OIDF-Pickering emulsions. (**A**) OIDF-Pickering emulsions stabilized by unmodified OIDF; (**B**) OIDF-Pickering emulsions stabilized by modified OIDF. All photographs were taken during 28 days of storage at room temperature.

**Figure 7 foods-10-02982-f007:**
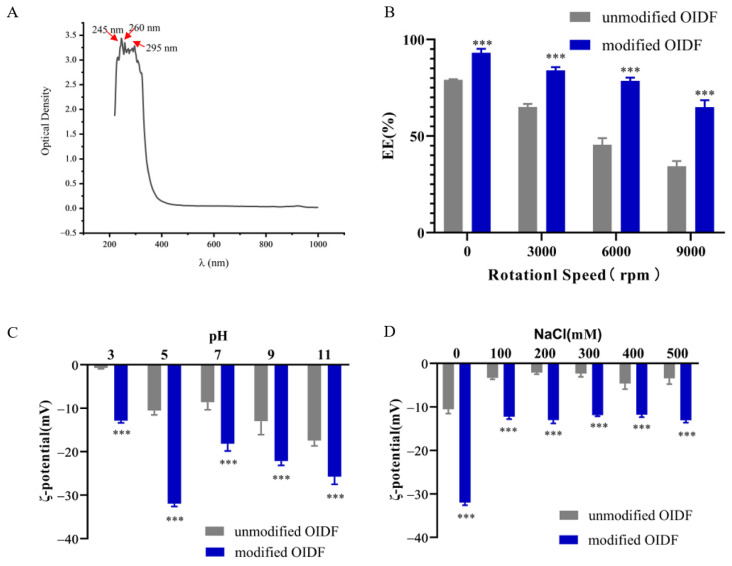
Stability of OIDF-Pickering emulsions. (**A**) The full wavelength scanning analysis of soybean oil; (**B**) the encapsulation efficiency (EE) of OIDF-Pickering emulsions at different centrifugal speeds; (**C**) The zeta potential of OIDF-Pickering emulsions at different pH levels after storage of 72 h; (**D**) The zeta potential of OIDF-Pickering emulsions with different NaCl addition after storage of 72 h (*** *p* < 0.001).

**Table 1 foods-10-02982-t001:** Optimization treatment parameters of OIDF-Pickering emulsion preparation.

Treatment	Soybean Oil (mL)	OIDF Suspensions (mL)
1	20	80 (1.0 wt%)
2	30	70 (1.0 wt%)
3	40	60 (1.0 wt%)
4	50	50 (1.0 wt%)
5	60	40 (1.0 wt%)
6	40	60 (0 wt%)
7	40	60 (0.4 wt%)
8	40	60 (0.8 wt%)
9	40	60 (1.2 wt%)
10	40	60 (1.6 wt%)
11	40	60 (2.0 wt%)

**Table 2 foods-10-02982-t002:** Changes in emulsified phase volume and emulsion feature of OIDF-Pickering emulsions during 28 days of storage at room temperature.

Sample	EPVF (%)					Emulsion Feature after 28 Days	
	0 Days	7 Days	14 Days	21 Days	28 Days	Upper Phase	Bottom Phase
UnmodifiedOIDF	100 ± 0.0	91.77 ± 4.6	87.01 ± 3.6	81.3 ± 2.2	78.75 ± 1.0	Gel-like emulsion	Aqueous phase
Modified OIDF	100 ± 0.0	100 ± 0.0	98.98 ± 2.3	97.44 ± 3.1	95.33 ± 3.9	Gel-like emulsion	Gel-like emulsion

## Data Availability

Raw data can be provided by the corresponding author on request.

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
