# Peer review of "Preparation and Characterization of Pickering Emulsions with Modified Okara Insoluble Dietary Fiber"

_foods, 2021, doi:10.3390/foods10122982_

Round 1

Reviewer 1 Report

This work evaluated the potential of modified OIDF as a Pickering emulsifier and the formation and stability of OIDF-Pickering emulsions. The topic is very interesting and of value to the scientific community. However, the article still needs some major revisions to improve readability, it is recommended that the article is also sent for an English check.

In the introduction authors should make it cleared what was the ultimate purpose of this study as in the abstract they say „Modified okara insoluble dietary fiber (OIDF) has been proved to be a Pickering emulsifier“. So if this already proven what is the added value of the present work?

In section 2.2. authors should provide more details on okara fermentation

Section 2.3 is very confusing and should be rewritten for better understanding of what was done. At one point authors write that the ultrasound treatment time was 3, 6, 9, 12, 15 min but then write that it was for 30 minutes. Which is the used treatment time ? If 30 minutes weren't the samples overheated? Did authors measure temperature increase during the treatment? Maybe if authors add all the tried formulation/treatment parameters in a table it would improve clarity.

In Line 125 what do authors mean by „type of emulsions was determined according to the principle of similar dissolution“?

Why did authors call their analysis Environmental stability? Is this really the environmental stability? I would rather say it is the response of emulsions to a pH variation. Environmental would have to be many other factors such as temperature, humidity, etc.

In results and discussion it is unclear if authors are talking about their own results or literature results as they provide references for most of the things (e.g. line 193, 195). Quality of Figure 1,2,and  5 must be improved as it is not possible to read what is written.

Some other minor comments:

Line 31:proteins, polysaccharides

Line 75: were purchased

Line 84: our  previous  method

Line 87: which dryer? Provide details

Line 92 and 93: Provide the intermediate ratios used

Line 110: electric potential dish? Was it not a cell?

Line 128, 135:  English check

Line 141: flat sheet die ? Is it not a flat plate?

Line 153: were placed

Line 165: Provide values in g

Line 418: remove recently

Author Response

Response to Reviewer 1 Comments

Dear reviewer,

Thank you very much for your attention and the referee’s evaluation and comments on our manuscript ID: foods-1448123 titled Preparation and characterization of Pickering emulsions with modified okara insoluble dietary fiberWe have revised the manuscript according to your kind advices and referee’s detailed suggestions. The specific parts are listed below.

Point 1: In the introduction authors should make it cleared what was the ultimate purpose of this study as in the abstract they say“Modified okara insoluble dietary fiber (OIDF) has been proved to be a Pickering emulsifier”. So if this already proven what is the added value of the present work?

Response 1: In the introduction, we have made some modifications.

Therefore, in this study preparation of Pickering emulsions using OIDF modified by yeast K. marxianus were compared with unmodified OIDF, and the stability of those OIDF-Pickering emulsions were also characterized. For the first time, the OIDF modified by microbial fermentation was applied to the field of Pickering emulsions. (Line 71).

Point 2: In section 2.2. authors should provide more details on okara fermentation

Response 2: In section 2.2, we have provided all the details on okara fermentation.

The okara was soaked in distilled water at a ratio of 1:5 (w/v) and sterilized at 121 °C for 20 min in a vertical pressure steam sterilization pot (YXQ-S-50A, Shanghai Boxun Enterprise Co., Ltd., Shanghai,China). The sterilized okara was inoculated with K. marxianus powder at a ratio of 10% (w/v) and fermented for 72 h.  (Line 87).

Point 3: Section 2.3 is very confusing and should be rewritten for better understanding of what was done. At one point authors write that the ultrasound treatment time was 3, 6, 9, 12, 15 min but then write that it was for 30 minutes. Which is the used treatment time ? If 30 minutes weren't the samples overheated? Did authors measure temperature increase during the treatment? Maybe if authors add all the tried formulation/treatment parameters in a table it would improve clarity.

Response 3: As for the method described in section 2.3.

The OIDF suspensions containing 0, 0.4, 0.8, 1.0, 1.2, 1.6, 2.0 wt% OIDF powder were mixed by a homogenizer (Tissue-Tearor, BioSpec, USA) at 10,000 rpm, for three times and each time for 3 min, and then operated with ultrasonic cell grinder (Ningbo Xinzhi Biotechnology Co., Ltd., Ningbo, China) at an ice bath at 500 W for 3 s and an interval of 3 s for 30 min.

The OIDF-Pickering emulsions, containing soybean oil and OIDF suspensions were emulsified according to the two-step method (shear and ultrasonication) proposed by Bai et al [20]. Firstly, OIDF-Pickering emulsions were homogenized by a homogenizer (Fluke Fluid Machinery Manufacturing Co., Ltd., America) at 13,000 rpm for 2 min, then immediately operated with ultrasonic cell grinder (Ningbo Xinzhi Biotechnology Co., Ltd., Ningbo, China) at 500 W for 3 s and an interval of 3 s for 3 min. The optimization of OIDF-Pickering emulsion preparation formulation was achieved by varying the ratio of soybean oil and OIDF suspensions specifically produced 11 different treatment combinations (Table. 1). (Line103).

Table 1. Optimization of OIDF-Pickering emulsion preparation treatment parameters

Treatment

Soybean oil (mL)

OIDF suspensions (mL)

1

20

80 (1.0 wt%)

2

30

70 (1.0 wt%)

3

40

60 (1.0 wt%)

4

50

50 (1.0 wt%)

5

60

40 (1.0 wt%)

6

40

60 (0 wt%)

7

40

60 (0.4 wt%)

8

40

60 (0.8 wt%)

9

40

60 (1.2 wt%)

10

40

60 (1.6 wt%)

11

40

60 (2.0 wt%)

Point 4: In Line 125 what do authors mean by“of emulsions was determined according to the principle of similar dissolution”?

Response 4: We have described the rule name and supplemented the reference.

Before that, the type of emulsions was determined according to the principle of “like dissolves like" (Line 139).

“Characterization of surimi particles stabilized novel pickering emulsions: Effect of particles concentration, pH and NaCl levels” doi:10.1016/j.foodhyd.2021.106731

Point 5: Why did authors call their analysis Environmental stability? Is this really the environmental stability? I would rather say it is the response of emulsions to a pH variation. Environmental would have to be many other factors such as temperature, humidity, etc.

Response 5: We revised it after discussion.

3.4.3. pH or ionic strength (NaCl) stability

The pH and ionic strength (NaCl) are elements in the emulsion preparation and application process. (Line 428).

Point 6: In results and discussion it is unclear if authors are talking about their own results or literature results as they provide references for most of the things (e.g. line 193, 195). Quality of Figure 1,2,and 5 must be improved as it is not possible to read what is written

Response 6: In the results and discussion, the reason why we provide references is to explain the significance and purpose of specific experiments, which we have revised here.

In addition, the formation and stability of Pickering emulsion are closely related to the morphology and properties of solid particles [24] (Line 207).

According to the characteristics of emulsion formation, the oil water ratio and the change of solid particles fulfil a crucial role in the size of emulsion droplets and the stability of emulsions [15]. (Line 235).

Figure 1、2 and 5 were remade.

About some other minor comments:

Answers: Regarding the small comments, we have revised them one by one, as follows.

Some other minor comments:

Answers:

Line 31:proteins, polysaccharides

Line 33

Line 75: were purchased

Line 79

Line 84: our previous method

Line 88

Line 87: which dryer? Provide details

Line 95

Line 92 and 93: Provide the intermediate ratios used

Line 103

Line 110: electric potential dish? Was it not a cell?

Line 123

Line 128, 135:  English check

Line 141 or 148

Line 141: flat sheet die ? Is it not a flat plate?

Line 155

Line 153: were placed

Line 167

Line 165: Provide values in g

Line 176 or 407

Line 418: remove recently

Line 442

 Many grammatical or comments errors have been revised. All the lines indicated above are in the revised manuscript. All the lines indicated above are in the revised manuscript. Thank you for your kind suggestions.

Sincerely yours,

Chunhong Piao

Professor

Jilin Agricultural University, China.

[email protected].

Reviewer 2 Report

Some comments are:

  • Abstract: Indicate the most relevant aspect of the work 
  • Include referencefor line "Pickering emulsion is a novel emulsion stabilized by amphiphilic solid particles
  • Include reference for line "Soybean ..... extract protein"
  • Include reference for line"Every year.... costs"
  • Is it the first work about this topic? Mention it
  • Mention how the OIDF chemical composition were obtained
  • Improve quality of figure 1A. It is difficult to observe
  • Improve quality of figure 1C. It is difficult to read. Include name of the axis X
  • Figure 2 has a lot of the information. Consider divided 
  • Compare the results of the figure 2 with comparable works 
  • Item 3.3: Check the line spacing
  • Improve quality of figure 3B.  Include name of the axis X
  • Improve quality of figure 5. It is difficult to read. Include name of the axis X 

Author Response

Response to Reviewer 2 Comments

Dear reviewer,

Thank you very much for your attention and the referee’s evaluation and comments on our manuscript ID: foods-1448123 titled Preparation and characterization of Pickering emulsions with modified okara insoluble dietary fiberWe have revised the manuscript according to your kind advices and referee’s detailed suggestions. The specific parts are listed below.

Point 1: Abstract: Indicate the most relevant aspect of the work.

Response 1: We have revised the abstract section. The most relevant aspect of this work is the preparation of Pickering emulsifier with oidf modified by yeast Kluyveromyces marxinus, and its formation and stability were characterized

Modified okara insoluble dietary fiber (OIDF) is attracted great interest as a promising Pickering emulsifier. At present, the modification methods are mainly physicochemical methods, and the research on microbial modified OIDF as stabilizer is not clear. In this work, modified OIDF was prepared by yeast Kluyveromyces marxianus fermentation. The potential of modified OIDF as a Pickering emulsifier and the formation and stability of OIDF-Pickering emulsions stabilized by modified OIDF were characterized, respectively. (Line 10).

Point 2: Include referencefor line "Pickering emulsion is a novel emulsion stabilized by amphiphilic solid particles.

Response 2: We have provided references.

Pickering emulsion is a novel emulsion stabilized by amphiphilic solid particles [1]. (Line 28).

“Characteristics of starch-based Pickering emulsions from the interface perspective”. doi:10.1016/j.tifs.2020.09.026.

Point 3: Include reference for line "Soybean ..... extract protein".

Response 3: We have provided references.

Soybean (Glycine max(Linn.)Merr.)is one of the important grain crops worldwide, which is commonly used to make various soybean products, extract soybean oil, brew soy sauce and extract protein [12]. (Line 45).

“Okara: A soybean by-product as an alternative to enrich vegetable paste”. doi:10.1016/j.lwt.2018.02.058.

Point 4: Include reference for line"Every year.... costs".

Response 4: We have provided references.

Every year, about 70 million tons of okara is obtained after soybean processing, which is used as animal feed or discarded as waste due to additional processing costs [13] (Line 47).

“Biovalorisation of okara (soybean residue) for food and nutrition”. doi:10.1016/j.tifs.2016.04.011.

Point 5: Is it the first work about this topic? Mention it.

Response 5: Yes, this is the first work on this topic.

Therefore, in this work, OIDF modified by microbial fermentation was applied to the field of Pickering emulsions for the first time. (Line 71).

Point 6: Mention how the OIDF chemical composition were obtained.

Response 6: We revised and referred to the method for the chemical composition of OIDF

The content of IDF in raw and modified OIDF were determined to 70.53% and 72.38% according to national standards GB 5009.88-2014, and the content of crude protein, crude lipid, moisture and ash in raw and modified OIDF were determined to 70.53%, 12.0%, 5.42%, 8.60%, 1.31% and 72.38%, 9.90%, 8.41%, 5.56%, 1.02% according to national standards GB 5009-2016. (Line 98).

Point 7: Improve quality of figure 1A. It is difficult to observe

Response 7: Figure 1A was redrawn.

Point 8: Improve quality of figure 1C. It is difficult to read. Include name of the axis X.

Response 8: Figure 1C was redrawn and the axis X name was provided.

Point 9: Figure 2 has a lot of the information. Consider divided.

Response 9: After our discussion, we believe that the information in Figure 2 has strong correlation and cannot be separated, but we have improved the quality of Figure 2 for viewing

Point 10: Compare the results of the figure 2 with comparable works.

Response 10: We provided content to compare the results of Figure 2 with comparable works

In similar studies, Cai et al., [16] used soybean dregs insoluble dietary fiber at a concentration of 1.0 wt% to stabilize the O/W emulsions, and Bai et al., [30] used a concentration of 2.0 wt% nanocellulose to stabilize the O/W emulsions. (Line 276).

Point 11: Item 3.3: Check the line spacing.

Response 11: We checked the line spacing of item 3.3 (Line 289).

Point 12: Improve quality of figure 3B. Include name of the axis X.

Response 12: Figure 3B was redrawn and the axis X name was provided.

Point 13: Improve quality of figure 5. It is difficult to read. Include name of the axis X.

Response 13: Figure 5 was redrawn and the axis X name was provided.

Many grammatical or comments errors have been revised. All the lines indicated above are in the revised manuscript. All the lines indicated above are in the revised manuscript. Thank you for your kind suggestions.

Sincerely yours,

Chunhong Piao

Professor

Jilin Agricultural University, China.

[email protected].

Round 2

Reviewer 1 Report

The article was considerably improved, I just have few other minor comments:

Line 10: has

Line 66: was: There is no need to repeat the results again

Line 106: Also mention the interval between these 3 minutes

Figure: Explain what is the meaning of „***“ in C

Line 302: similar

Figure 4: it is not clear what the difference between A1 and A2 is, is it just repetition of same formulation or different formulation? If same formulation I would just leave one image to avoid confusion

Line 366: remove stronger stronger

Figure 7: remove environmental stability from title and explain what is the meaning of „***“

Author Response

Response to Reviewer 1 Comments Dear reviewer, Thank you very much for your attention and comments on our manuscript ID: foods-1448123 titled “Preparation and characterization of Pickering emulsions with modified okara insoluble dietary fiber” as well as your recognition of our last revision work. We have revised your latest comments and suggestions. The specific part is as follows. Point 1: Line 10: has. Response 1: “Is” has been changed to “has” specifically in line 10. Modified okara insoluble dietary fiber (OIDF) has attracted great interest as a promising Pickering emulsifier. (Line 10). Point 2: Line 66: was: There is no need to repeat the results again. Response 2: “Is” has been changed to “was” specifically in line 66. It was found that the modified OIDF exhibited (Line 66). Point 3: Line 106: Also mention the interval between these 3 minutes Response 3: The interval indicated in line 106 is 30 s, and the specific revision is as follows. The OIDF suspensions containing 0, 0.4, 0.8, 1.0, 1.2, 1.6, 2.0 wt% OIDF powder were mixed by a homogenizer (Tissue-Tearor, BioSpec, USA) at 10,000 rpm, for three times and each time for 3 min, each interval is 30 s (Line 101). Point 4:Figure: Explain what is the meaning of “***” in C Response 4: In section 2.10 of the manuscript, we provide a minimum significance value of 0.05, so here we specifically explain the meaning of ***, as followsand the specific revision is as follows. (*p < 0.05,**p < 0.01,***p < 0.001). (Line 229). Point 5: Line 302: similar Explain what is the meaning of “***” Response 5: Same as the point 4. (*p < 0.05,**p < 0.01,***p < 0.001). (Line 302). Point 6: Figure 4: it is not clear what the difference between A1 and A2 is, is it just repetition of same formulation or different formulation? If same formulation I would just leave one image to avoid confusion Response 6: In the Figure .4, A1 indicates the OIDF-Pickering emulsions stabilized by unmodified OIDF, and A2 indicates the OIDF-Pickering emulsions stabilized by modified OIDF. (Line 325) Point 7: Line 366: remove stronger stronger Response 7: After removing “stronger stronger,” the details are as follows. The consequence is mainly attributed to higher porosity, resulting in dispersion of re-sistance to shear stress and stronger, and stronger electrostatic repulsion force between modified OIDF, resulting in more stability of network structure (Line 358). Point 8: Figure 7: remove environmental stability from title and explain what is the meaning of “***” Response 8: Same as the point 4. (*p < 0.05,**p < 0.01,***p < 0.001) (Line 403). Many grammatical or comments errors have been revised. All the lines indicated above are in the revised manuscript. Thank you for your kind suggestions. Sincerely yours, Chunhong Piao Professor Jilin Agricultural University, China. [email protected].
